# Medical ontology learning framework to investigate daytime impairment in insomnia disorder and treatment effects

Alexander J. Büsser [1] ✉, Renato Durrer[2], Moritz Freidank[2], Matteo Togninalli[2], Antonio Olivieri[1], Michael A. Grandner[3,5] & William V. McCall [4,5]

## Abstract

**Background** Specificity challenges frequently arise in medical ontology used for the representation of real-world data, particularly in defining mental health disorders within widely used classification systems such as the International Classification of Diseases (ICD). This study aims to address these challenges by introducing the Disease-Specific Medical Ontology Learning (DiSMOL) framework, designed to generate precise disease representations from clinical physician notes, with a focus on daytime impairment in insomnia disorder.

**Methods** The study applied the Disease-Specific Medical Ontology Learning framework to clinical notes to better represent daytime impairment. The framework's performance was compared to insomnia expert-selected codes from ICD. Key statistical methods included sensitivity and F1-score comparisons, as well as analysis of symptom changes after the use of various medications, including benzodiazepines, non-benzodiazepine receptor agonists, and trazodone.

**Results** The DiSMOL framework significantly enhances the identification of daytime impairment in people with insomnia. Sensitivity increases from 17% to 98%, and the F1-score improves from 28% to 86%, compared with expert-selected ICD codes. Additionally, the framework reveals significant increases in daytime impairment symptoms following benzodiazepine use (18.9%), while traditional ICD codes do not detect any significant change.

**Conclusions** The study demonstrates that DiSMOL offers a more accurate method for identifying specific disease aspects, such as daytime impairment in insomnia, than traditional coding systems. These findings highlight the potential of specialized ontologies to enhance the representation and analysis of real-world clinical data, with important implications for healthcare policy and personalized medicine.

## Plain language summary

This study introduces an approach to improve the identification of specific symptoms seen in people with mental health conditions, particularly insomnia, from medical records. Traditional classification systems often fail to accurately capture issues such as inability to function during the daytime. This can make it challenging to properly assess and compare the effectiveness of treatments given to people that aim to improve their mental health. We developed a computational framework that can more precisely analyze medical records to identify these symptoms. Our computational method outperformed traditional approaches, offering a more accurate way to detect daytime impairment in people with insomnia. Using our approach could enable more accurate research into insomnia. Also, this could potentially accelerate the development and accurate evaluation of treatments for insomnia, and mental health conditions leading to better outcomes for patients.

In medicine, ontology refers to the formal representation of medical concepts to enable clear and unambiguous communication through defined medical terms, often using the International Classification of Diseases (ICD) codes. The development of medical ontologies is rigorous and lengthy, involving the collaboration of knowledge engineers and physicians[1]. Despite these efforts, medical ontologies often lack specificity for a particular disease and are influenced by administrative processes[2,3]. Recent advancements in natural language processing and the availability of large amounts of semantically rich medical information have given rise to the field of ontology learning. Ontology learning involves the semi-automatic identification and extraction of key concepts from text to enable the construction of an ontology[4], and can increase the accuracy of disease representations[5].

Real-world data is now widely used to assess disease burden, treatment effectiveness, and safety. Unlike controlled clinical trials, researchers lack

[1]Idorsia Pharmaceuticals, Allschwil, Switzerland. [2]Visium, Prilly, Switzerland. [3]Department of Psychiatry, University of Arizona College of Medicine, Tucson, AZ, USA. [4]Department of Psychiatry and Health Behavior, Medical College of Georgia at Augusta University, Augusta, GA, USA. [5]These authors jointly supervised this work: Michael A. Grandner, William V. McCall. ✉e-mail: aj.buesser@gmail.com

control and guidance over data reporting in real-world studies. Furthermore, real-world evidence studies often suffer from a lack of transparency regarding the reliability of selected ontologies, and a more rigorous assessment of the ontologies utilized is warranted. Here, we introduce the Disease-Specific Medical Ontology Learning (DiSMOL) framework, an innovative approach to ontology learning where pre-existing ontologies are enriched with real-world data and carefully evaluated. DiSMOL uses physician notes to extend medical ontologies, providing first-hand access to real-world reporting practices and language use among healthcare providers. This approach involves elements of deep phenotyping, particularly in systematically characterizing and learning about disease symptoms. However, it is also crucial to recognize that our work extends beyond mere symptom documentation. By incorporating these symptoms into an existing knowledge base, we essentially engage in a form of ontology learning, expanding and refining this knowledge base, which already has a specific structural framework.

To demonstrate the potential of DiSMOL, we applied the framework to examine the burden of daytime impairment in people with insomnia disorder, the most prevalent worldwide sleeping disorder. Daytime impairment is a required feature of insomnia diagnosis[6] and a very relevant aspect to be addressed for individuals affected by the disease. Improving daytime functioning has been identified as the most important attribute of an insomnia treatment by people with insomnia disorder[7]. Despite its significance and importance for individuals with insomnia, symptoms describing daytime impairment are rarely coded. Therefore, DiSMOL aims to identify as many genuine daytime impairment symptoms as possible while minimizing the inclusion of spurious symptoms to provide an accurate representation of this critical aspect of insomnia disorder. In this paper, we show that DiSMOL is able to detect symptoms of daytime impairment in people with insomnia with greater sensitivity compared to ICD-10 codes. Additionally, the framework reveals significant increases in daytime impairment symptoms following benzodiazepine use (18.9%), while traditional ICD codes do not detect any significant change.

## Methods

### Data source and pre-processing
To develop the disease-specific language model, we accessed data on 1.9 million patients from the HealthVerity longitudinal repository. This included medical insurance claims data from an undisclosed provider, represented as ICD-10 codes, and physician notes in free-text form from Amazing Charts LLC. In compliance with HIPAA regulations, all data used in this study was de-identified, containing no personally identifiable information. As such, institutional review board (IRB) approval was not required for the initial setup of the database and the resulting study data, as the analysis of de-identified data does not constitute human subjects' research. Likewise, informed consent was not necessary due to the de-identified nature of the dataset.

Of the 1.9 million patients identified, data from 82,722 patients diagnosed with insomnia (Supplementary Table 1) and at least one physician note (486,885 visit notes in total) were considered and split into a model training set of 82,672 patients and a test set of 50 patients. Both sets were stratified by age and sex. Only information on the chief complaint, review of the system, physical exam, the assessment, and history of present illness were considered for model training. Data related to past medical history was discarded, as it may have contained information on diseases not related to insomnia.

### Using a Simple Ontology as Model Input
The Insomnia Daytime Symptoms and Impacts Questionnaire (IDSIQ)[8] was used as a starting point to define the disease-specific language model input (referred to as the simple ontology). IDSIQ is a patient-reported outcome instrument that consists of 14 items, an item being a sentence describing a daytime impairment symptom as perceived by patients, which were grouped into three domains (cognition, emotional, physical) by the insomnia experts (Supplementary Methods Insomnia Experts). While a

different instrument or knowledge base could have been considered, the IDSIQ was chosen by insomnia experts because of its totality in describing symptoms of daytime impairment. To complement the patient-reported symptoms, two insomnia experts provided terms of daytime impairment commonly used in their clinical practice to describe the symptoms reported by the patients. While words such as sleepy, tired, drowsy, and fatigue[9] have different meanings to patients, the same cannot be assumed of clinicians who are not experts in insomnia. To account for the potential heterogeneous language used by non-sleep specialists, the terms provided by the insomnia experts - along with the original IDSIQ items - were used as input (referred to as the simple ontology) to find contextually similar words through the DiSMOL framework (Fig. 1, Supplementary Data 1).

### Learning a disease specific model
DiSMOL used Word2vec[10] and hyperparameter optimization. The hyperparameter space (Supplementary Table 3) was restricted to a recommended range[11], and the models were optimized using a metric inspired by Beam and colleagues[12]. The metric involved creating a list of insomnia symptom pairs with a known relationship (Supplementary Data 4), assigning a semantic type to each token, randomly drawing 10,000 tokens of the semantic supertype, daytime impairment (Supplementary Table 5), and calculating the cosine similarity for each pair of tokens. The resulting distribution of cosine similarities was used to evaluate the models, with the metric being the fraction of known relationships within the 95th percentile cosine distance distribution.

Eight models, trained on different bootstrap samples each using the best performing hyperparameter configuration, were used to create an ensemble model. This involved averaging distances between tokens across the eight models to reduce variations in close neighbors due to small changes in the data and initialize the weights of the model. To assess the suitability of pre-trained language models versus disease-specific models, popular models such as Bidirectional Encoder Representations from Transformers (BERT) were evaluated alongside the disease-specific Word2Vec ensemble model. The disease-specific model (Supplementary Table 2) outperformed BERT, in median performance across 5 independent runs by 83% (Table 1), enhancing the performance for downstream tasks.

### Ontology enrichment through disease specific model
The simple ontology was subsequently enriched by retrieving additional terms from the disease-specific language model. Only pairs (x: an item or corresponding insomnia expert synonym, y: retrieved word2vec term) within the 0.963 percentile with respect to the distribution of cosine distances were retrieved, and only those with the semantic supertype of daytime impairment were kept (Supplementary Table 5). Furthermore, the words needed to appear in at least 0.24% of all patients and have a clear semantic directionality as determined by the medspaCy ConText algorithm[13]. The remaining set of 425 terms was further investigated by insomnia experts and words not describing daytime impairment were removed resulting in 109 words (Supplementary Data 2). We acknowledge that the validation rate of these terms was approximately 25%. Regarding the terms that were ultimately discarded, our analysis shows that while they did not meet the criteria for inclusion in the final dataset, many were loosely related to the domain of interest but lacked the necessary specificity or direct relevance to daytime impairment as defined for our study. To clarify, the system was not designed for fully automated use; instead, it was developed to aid experts by providing a broader array of potential terms for further scrutiny. Finally, the daytime impairment ontology was examined for lexical consistency (i.e., words with the same lemma were present in all or no domains).

By following the DiSMOL approach, the language of insomnia experts was enriched by incorporating terminology used by a large sample of healthcare providers and non-sleep experts, without needing to conduct an extensive and time-consuming questionnaire. Furthermore, as this approach uses physician notes written by healthcare providers prior to using these notes for research purpose, there is no potential bias with respect to the

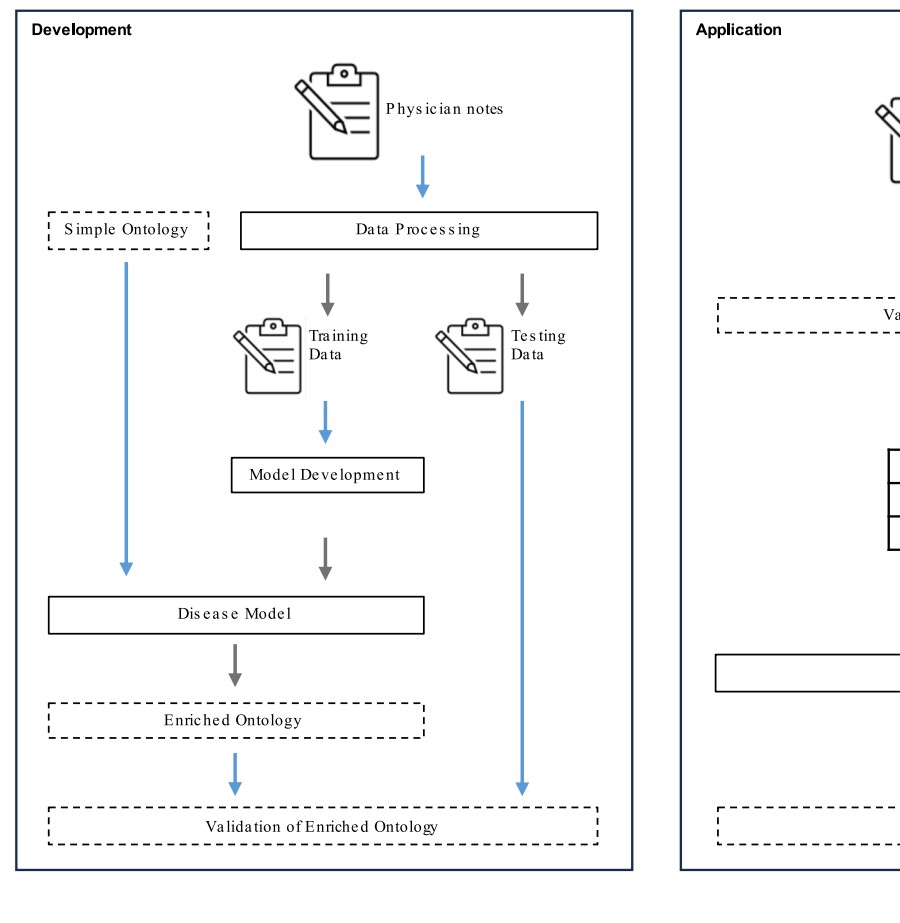

**Fig. 1 | Disease-Specific Medical Ontology Learning (DiSMOL) Framework.** The disease-specific language model is trained on physician notes. Known symptoms, as informed by insomnia experts or existing instruments form the simple ontology, which is used as an input to the model to find contextually similar words. The resulting enriched ontology is then validated by an expert panel. Subsequently, the validated enriched ontology is utilized to transform physician notes into structured data, enabling the investigation of the disease burden.

Hawthorne Effect and therefore provides a representation of language use in everyday clinical practice. Lastly, owing to minimal human intervention (Fig. 1), the approach highlights scalability and applicability to other disorders.

### Inference of the daytime impairment ontology on text data

The ontology was utilized to identify terms from physician notes, representing the presence of a daytime impairment for a patient at a specific encounter. The medspaCy ConText algorithm was utilized to measure the correct semantic directionality of terms. The presence of a daytime impairment symptom was only confirmed if a word had the correct semantic directionality. For instance, "being happy" would not constitute an impairment, but "not being happy" would be identified as one. The presence of an impairment was assessed on the level of a patient encounter with a healthcare provider. An encounter was considered to have an impairment if one or more daytime impairment symptoms were present. The number of impairments for a specific encounter was not considered.

### Ontology evaluation

The accuracy of the enriched insomnia daytime impairment ontology was evaluated against two baseline representations based on ICD-10 codes. The first representation (ICD-clin) used a set of ICD-10 codes selected by insomnia experts (Supplementary Table 4). For the second representation (ICD-clin-DiSMOL), insomnia experts matched ICD-10 codes with symptoms generated from DiSMOL framework (Supplementary Data 3).

To generate our ground truth data, an insomnia expert annotated 108 physician notes from the test set of 50 patients according to the presence or absence of physician-reported daytime impairment symptoms in the three domains: cognition, emotional, and physical. The ability of ICD-clin, ICD-clin-DiSMOL, and DiSMOL to match ground truth data was evaluated using the sensitivity, precision, and F1-score[14].

### Treatment effect estimation

To compare the ability of ICD-clin, ICD-clin-DiSMOL, and DiSMOL to quantify the effect of treatment on the number of daytime impairment symptoms in individuals with insomnia, three distinct patient populations were identified based on the treatment administered: trazodone, benzodiazepine consisting of estazolam, flurazepam, lorazepam, quazepam, temazepam, triazolam), and non-benzodiazepine receptor agonists (non-BzRAs) consisting of eszopiclone, zaleplon, zolpidem. Of the 82,722 patients originally identified, only those with an insomnia diagnosis within 6 months prior to treatment start and aged ≥18 years at baseline were included. Exclusion criteria included an insomnia diagnosis date >6 months before treatment start, palliative care, active malignancies, or pregnancy. Patients were followed retrospectively for 182 days (6 months) prior to their first pharmacological treatment for insomnia and for an average of three months during treatment. To estimate treatment effect, the within-subject relative difference in the number of daytime impairment symptoms prior to and during treatment with trazodone ($n = 1522$ patients), benzodiazepine ($n = 1045$ patients), and non-BzRAs ($n = 2361$ patients) were measured. A

censoring of the treatment start day was applied for the calculation of the treatment effects. The statistical significance of the difference in outcome between the pre-treatment period and post-treatment period was calculated using a two-sided t-test. Multiple hypothesis tests using the false discovery rate (FDR) corrections with the Benjamini–Hochberg procedure[15] were applied. The effect estimation was calculated by counting the number of reported symptoms in 100 patient-years and subtracting the number of symptoms in the baseline period reported in 100 patient-years. CIs <0.05 were considered statistically significant.

Although the baseline characteristics of the three study populations were comparable, except for psychiatric co-morbidities, which were less frequent in the non-BzRA group (Table 2), the study design did not allow for a comparison between the different treatments, as the control for confounders only applies within a specific population.

## Statistics and reproducibility
All statistical tests used, the sample sizes, the number of replicates, and how replicates were defined are described in the methods section under data source and preprocessing, learning a disease specific model, ontology enrichment through disease specific model, ontology evaluation, and treatment effect estimation.

## Reporting summary
Further information on research design is available in the Nature Portfolio Reporting Summary linked to this article.

# Results
## Daytime impairment ontology
As described in the Methods, we compared the accuracy of the enriched insomnia daytime impairment ontology against two baseline representations based on ICD-10 codes. The enriched ontology was used to transform

physician notes into structured data to identify the presence of daytime impairment symptoms during patient visits (see Fig. 1). As expected, due to ambiguity in human language, DiSMOL uncovered a noteworthy overlap of daytime impairment symptoms between the physical domain and the cognition and emotional domains (Fig. 2). Symptoms like "dizzy", "tiredness", "weak", "somnolence", and "sleepiness" are shared between the physical and cognition domains, while terms such as "depressed," "depression," and "fatigued" are shared between the physical and emotional domains. Interestingly, "fatigued" stands out as the sole identified symptom that relates to all three domains, serving as the link between the emotional domain and symptoms of cognition.

## Ontology evaluation
While the precision of the ontologies exhibited a similar performance, DiSMOL displayed a substantial increase in sensitivity, improving by 476% from 17% to 98%, and a noteworthy enhancement of the F1-score by 207%, rising from 28% to 86% when compared with ICD-clin. An increase in sensitivity, rising from 17% to 51%, when utilizing ICD-clin-DiSMOL in comparison to ICD-clin, implies that insomnia experts can enhance their code selection by incorporating a broader range of symptoms, as indicated by DiSMOL (Fig. 3). An indication of reduced sensitivity in the ICD benchmarks implies that daytime impairment may be underreported in insurance claims. In this context, DiSMOL stands out as the most accurate method for measuring daytime impairment symptoms. Interestingly the inclusion of patient relevant symptoms did not harm the precision of DiSMOL compared with ICD-clin. This may suggest that symptoms that are relevant to patients are also relevant to clinicians, an observation which warrants further investigations.

## Treatment effect estimation on daytime impairment
In the absence of randomized clinical trials, we investigated and compared the ability of ICD-clin, ICD-clin-DiSMOL, and DiSMOL to quantify the effect of treatment with trazodone, benzodiazepine, and non-benzodiazepine receptor agonists (non-BzRAs) on the number of daytime impairment symptoms in individuals with insomnia. These hypnotics were selected as the most widely used sleep medications which also cause adverse events that might interfere with the daytime functioning of the patients. Melatonin was not included in the analysis as it is not always easy to understand which specific melatonin product is used and OTC formulations may not always be reported. Quantifying daytime impairment symptoms during versus before treatment with benzodiazepines, trazodone, and non-BzRAs differed between DiSMOL and ICD-clin. Compared with before treatment, treatment with benzodiazepines was associated with a statistically significant increase of daytime impairment symptoms in all

## Table 1 | Model results

| Approach | Fraction of Seed Pairs in 95th Percentile (Median Across 5 Runs)[1] |
|---|---|
| Disease specific model using a Word2vec 8-model ensemble | 0.42 |
| BioBERT (v1.1) | 0.23 |
| BERT (base-uncased) | 0.21 |
| BERT (large-uncased) | 0.07 |
| Bio-ClinicalBERT | 0.02 |

## Table 2 | Baseline characteristics of the study population

| Characteristic (ICD-10 code)[a] | Trazodone (n = 1522) | Non-benzodiazepine receptor agonists (n = 2361) | Benzodiazepine (n = 1045) |
|---|---|---|---|
| Mean age ± SD, in years | 48.1 ± 15.2 | 48.7 ± 13.5 | 50.9 ± 16.0 |
| Female sex | 59.4 | 57.6 | 59.6 |
| Arterial hypertension (I10–I15) | 46 | 42 | 45 |
| Diabetes Type II (E11) | 20 | 17 | 17 |
| Anxiety (F41) | 34 | 25 | 39 |
| Depression (F32, F33) | 33 | 20 | 28 |
| Psychiatric comorbidities (F32, F33, F41, F10–F19) | 48 | 35 | 48 |
| Obesity (E66) | 22 | 18 | 19 |
| Heart failure (I50) | 3 | 3 | 5 |
| Ischemic heart disease (I20–I24) | 8 | 7 | 9 |
| Chronic obstructive pulmonary disease (J44) | 8 | 4 | 7 |
| Cerebral infarction (I63) | 2 | 2 | 2 |

*ICD* International Classification of Diseases, *SD* standard deviation.
[a]Values are expressed as % unless otherwise indicated.

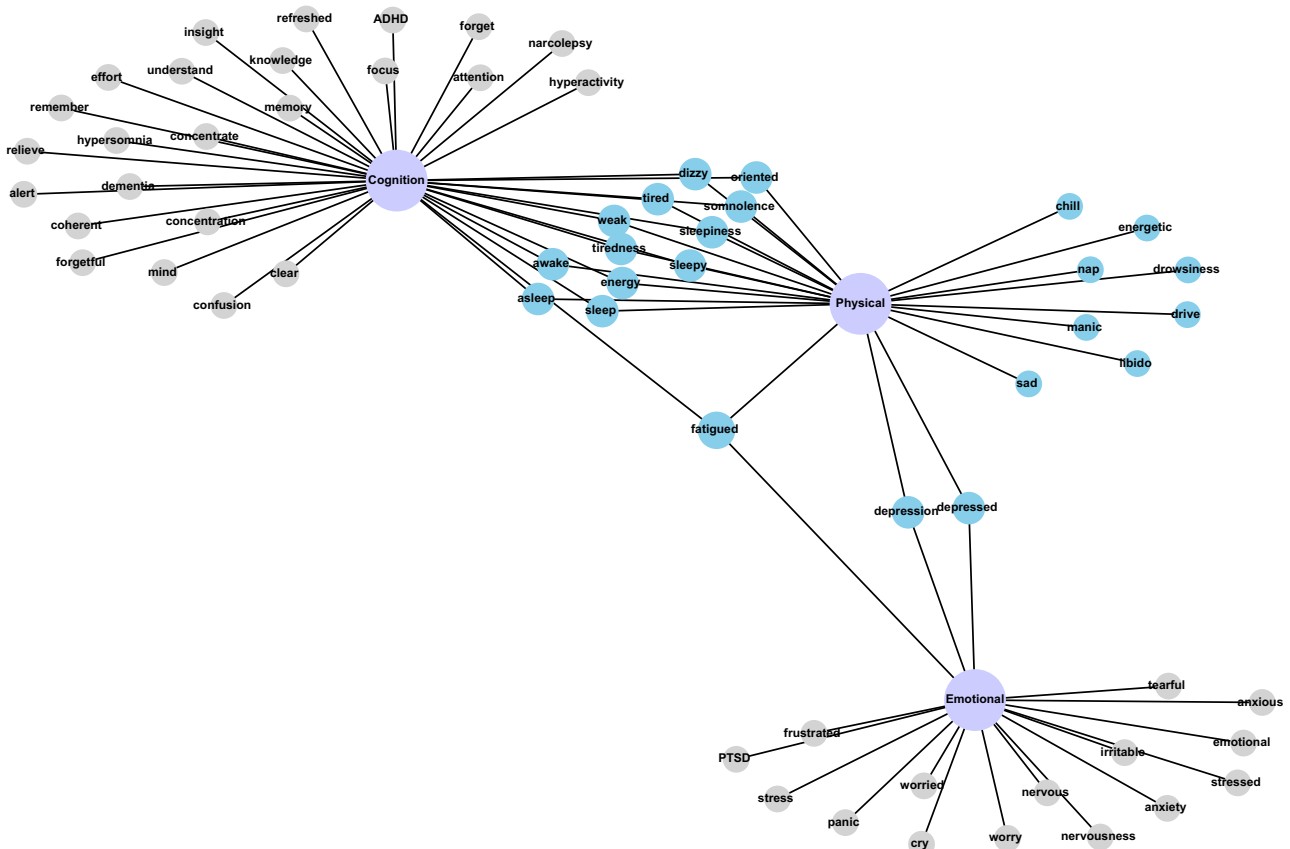

**Fig. 2 | Daytime impairment ontology.** A schematic representation of the validated enriched ontology highlighting the relationship between the daytime impairment symptoms and the domains colored in purple. Notably, symptoms linked with the physical domain are visually highlighted in light blue, serving to underscore their considerable overlap with the other domains. The cognition domain exhibits the largest array of associated symptoms, followed closely by the physical domain. Grey circles: DiSMOL synonym from the Cognition or Emotional domain; Purple circles: Domain names; Light blue circles: DiSMOL synonym from the Physical domain

domains ('overall') using DiSMOL (18.9%, $P = 0.021$), but not ICD-clin ($-3.3\%$, $P = 0.867$ [not significant (ns)]). Similar outcomes for DiSMOL versus ICD-clin were identified in the emotional (26.3%, $P = 0.016$ vs. $-13.2\%$, $P = 0.866$ [ns]) and physical (20.9%, $P = 0.032$ vs. $-11.1\%$, $P = 0.866$ [ns]) domains but not in the cognition domain (6.9%, $P = 0.793$ [ns] vs. $-3.3\%$, $P = 0.867$ [ns]) (Table 3). Patients who received trazodone experienced a statistically significant increase in daytime impairment symptoms overall during treatment compared with before treatment by ICD-clin (53.5%, $P = 0.014$). In comparison, no significant effect was observed for DiSMOL (9.0%, $P = 0.094$ [ns]) (Table 4). Similar outcomes for ICD-clin versus DiSMOL were identified for the cognition (53.5%, $P = 0.014$ vs. 4.4%, $P = 0.465$ [ns]), emotional (49.9%, $P = 0.018$ vs. 12.9%, $P = 0.071$ [ns]), and physical (50.1%, $P = 0.018$ vs. 7.0%, $P = 0.237$ [ns]) domains. Treatment with non-BzRAs compared with before treatment was associated with statistically significant increases across all representations in daytime impairment symptoms overall (DiSMOL: 16.0%, $P = 0.001$; ICD-clin: 43.5%, $P = 0.002$) and in each domain (cognition: 16.1%, $P = 0.003$ vs. 43.5%, $P = 0.002$; emotional: 21.6%, $P = 0.001$ vs. 43.0%, $P = 0.002$; physical: 11.8%, $P = 0.016$ vs. 43.2%, $P = 0.002$) (Table 5).

## Discussion

In our study, the DiSMOL framework showed an increase in daytime impairment symptoms for benzodiazepines and non-BzRAs. The use of ICD-10 codes only would have overestimated the daytime impairment associated with non-BzRAs and recorded a non-significant reduction in daytime impairment symptoms for benzodiazepines. Furthermore, the DiSMOL framework did not detect an increase in daytime impairment symptoms for trazodone, unlike the increase measured using ICD-10 codes. Interestingly, despite benzodiazepines being primarily indicated for mood

disorders such as anxiety and panic attacks, no improvement in the emotional domain of people with insomnia and the use of benzodiazepines was observed using DiSMOL. It is worth noting that insomnia can exacerbate anxiety and panic disorders, and conversely, anxiety and panic disorders can contribute to the development or perpetuation of insomnia[16]. Therefore, addressing one condition alone may not be sufficient to fully manage the interplay between the two, necessitating a more comprehensive treatment approach. Of note, benzodiazepines and non-BzRAs with varying half-lives were grouped together in this analysis, and consequently, the results observed for the entire class may differ for different drugs within this category.

This study introduces the DiSMOL framework, an innovative approach that enriches medical ontologies through real-world data and deep phenotyping. Our results suggest that disease representations learned from physician notes are better equipped to discriminate people with daytime impairment compared to ICD-10 codes selected by sleep experts. The DiSMOL framework boasts scalability, potential applicability across various disorders, and a commendable capability to precisely depict disease burden. However, this study does exhibit certain limitations. The validation process, led by insomnia experts, might introduce subjective biases as it is based on healthcare professionals' interpretation of patients' description of how they perceive the diseases. Furthermore, the validation was conducted on only 50 patients, which may limit the generalizability of the model's performance. It is important to acknowledge that the exclusive focus on insomnia disorder limits the framework's transferability to other medical conditions, necessitating further validation. The reliance on physician notes for ontology enrichment introduces challenges related to data quality and completeness. Moreover, design limitations preclude a direct contrast between different treatments. Time-varying variables and carry-over effects from prior

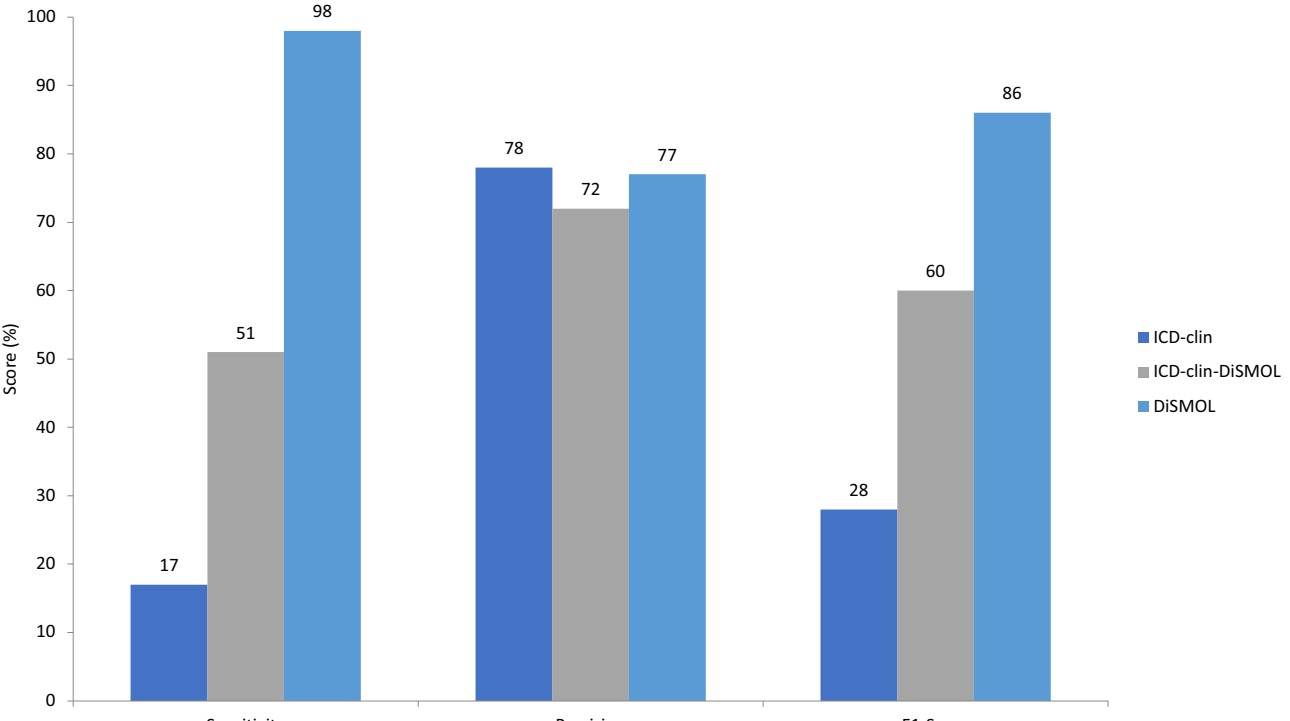

**Fig. 3 | Comparison of ontology performance.** This bar chart presents a comparison of three methods—ICD-clin, ICD-clin-DiSMOL, and DiSMOL—across three key performance metrics: Sensitivity, Precision, and F1-Score. DiSMOL demonstrates the highest performance in Sensitivity (98%) and F1-Score (86%), while ICD-clin leads slightly in Precision (78%). ICD-clin-DiSMOL shows moderate scores across all metrics, achieving 51% in Sensitivity, 72% in Precision, and 60% in F1-Score. The source data for Fig. 3 is in Supplementary Data 5. DiSMOL Disease-Specific Medical Ontology Learning, ICD International Classification of Diseases.

**Table 3 | The effect of benzodiazepines on daytime impairment symptoms by domain and ontology**

| Endpoint Ontology | Number of daytime impairment symptoms [events per 100 patient years][a] | | | Percentage increase in endpoint after treatment ± SD (P-value)[b] |
|---|---|---|---|---|
| | Before treatment | During treatment | Average treatment effect [95% CI] | |
| **All domains** | | | | |
| ICD-clin | 15.2 | 14.7 | −0.5 [−6.6, 5.6] | −3.3% ± 100.9% (P = 0.867 [ns]) |
| ICD-clin-DiSMOL | 67.6 | 89.6 | 22.0 [3.3, 40.8] | **32.6% ± 295.5% (P = 0.009)** |
| DiSMOL | 134.3 | 159.7 | 25.4 [-3.4, 54.2] | **18.9% ± 474.8% (P = 0.021)** |
| **Cognition domain** | | | | |
| ICD-clin | 15.2 | 14.7 | −0.5 [−6.6, 5.6] | −3.3% ± 100.9% (P = 0.867 [ns]) |
| ICD-clin-DiSMOL | 32.6 | 33.7 | 1.1 [−10.2, 12.4] | 3.3% ± 172.2% (P = 0.867 [ns]) |
| DiSMOL | 94.2 | 100.7 | 6.5 [−13.9, 26.8] | 6.9% ± 334.7% (P = 0.793 [ns]) |
| **Emotional domain** | | | | |
| ICD-clin | 13.2 | 11.5 | −1.7 [−7.4, 4.0] | −13.2% ± 93.9% (P = 0.866 [ns]) |
| ICD-clin-DiSMOL | 44.9 | 66.2 | 21.3 [5.6, 37.0] | **47.4% ± 251.3% (P = 0.004)** |
| DiSMOL | 83.5 | 105.4 | 22.0 [−2.0, 45.9] | **26.3% ± 394.5% (P = 0.016)** |
| **Physical domain** | | | | |
| ICD-clin | 13.8 | 12.3 | −1.5 [−7.2, 4.1] | −11.1% ± 92.8% (P = 0.866 [ns]) |
| ICD-clin-DiSMOL | 16.1 | 15.1 | −1.1 [−7.2, 5.1] | −6.6% ± 101.2% (P = 0.867 [ns]) |
| DiSMOL | 94.4 | 114.2 | 19.7 [−4.8, 44.3] | **20.9% ± 404.9% (P = 0.032)** |

The use of benzodiazepines is associated with a statistically significant increase in daytime impairment symptoms in the all domains, emotional domain, and physical domain categories when assessed using DiSMOL. In contrast, standard ICD-clin-based assessments showed no significant changes across any domain (P-values ≥ 0.867).
CI confidence interval, DiSMOL Disease-Specific Medical Ontology Learning, ICD International Classification of Diseases, ns not significant, SD standard deviation.
[a]The follow up period was from 182 days before treatment to approximately 3 months (mean) during treatment.
[b]An increase shows worsening of daytime functioning (more daytime impairment symptoms) after treatment compared with before treatment, while a decrease shows an improvement in daytime functioning (fewer daytime impairment symptoms).

**Table 4 | The effect of trazodone on daytime impairment symptoms by domain and ontology**

| Endpoint Ontology | Number of daytime impairment symptoms [events per 100 patient years][a] | | | Percentage increase in endpoint after treatment ± SD (*P*-value)[b] |
|---|---|---|---|---|
| | **Before treatment** | **During treatment** | **Average treatment effect [95% CI]** | |
| **All domains** | | | | |
| ICD-clin | 12.6 | 19.4 | 6.8 [−0.0, 13.5] | **53.5% ± 134.7% (*P* = 0.014)** |
| ICD-clin-DiSMOL | 69.6 | 91.8 | 16.4 [−1.3, 34.1] | **32.0% ± 351.9% (*P* < 0.001)** |
| DiSMOL | 137.7 | 150.1 | 12.4 [−7.4, 32.1] | 9.0% ± 392.7% (*P* = 0.094 [ns]) |
| **Cognition domain** | | | | |
| ICD-clin | 12.6 | 19.4 | 6.8 [−0.0, 13.5] | **53.5% ± 134.7% (*P* = 0.014)** |
| ICD-clin-DiSMOL | 34.9 | 43.4 | 6.4 [−3.6, 16.3] | **24.3% ± 198.5% (*P* = 0.037)** |
| DiSMOL | 95.1 | 99.4 | 4.2 [−10.2, 18.6] | 4.4% ± 286.1% (*P* = 0.465 [ns]) |
| **Emotional domain** | | | | |
| ICD-clin | 12.3 | 18.4 | 6.1 [−0.6, 12.8] | **49.9% ± 134.0% (*P* = 0.018)** |
| ICD-clin-DiSMOL | 43.0 | 58.0 | 9.2 [−5.6, 24.1] | **35.1% ± 295.4% (*P* = 0.002)** |
| DiSMOL | 87.1 | 98.3 | 11.2 [−4.8, 27.2] | 12.9% ± 317.6% (*P* = 0.071 [ns]) |
| **Physical domain** | | | | |
| ICD-clin | 12.5 | 18.8 | 6.3 [−0.5, 13.0] | **50.1% ± 134.4% (*P* = 0.018)** |
| ICD-clin-DiSMOL | 19.1 | 24.4 | 5.3 [−2.6, 13.2] | 27.8% ± 157.8% (*P* = 0.072 [ns]) |
| DiSMOL | 110.8 | 118.6 | 7.7 [−9.1, 24.6] | 7.0% ± 334.5% (*P* = 0.237 [ns]) |

Trazodone did not demonstrate statistically significant changes across any domain (*P*-values ≥ 0.071) when using the DiSMOL method. However, when assessed using ICD-clin and ICD-clin-DiSMOL, trazodone was associated with a statistically significant increase in daytime impairment symptoms across the all domains, cognition domain, emotional domain, and physical domain categories. Notably, the observed increase in daytime impairment may be overestimated when relying on ICD-clin and ICD-clin-DiSMOL.
*CI* confidence interval, *DiSMOL* Disease-Specific Medical Ontology Learning, *ICD* International Classification of Diseases, *ns* not significant, *SD* standard deviation.
[a]The follow up period was from 182 days before treatment to approximately 3 months (mean) during treatment.
[b]An increase shows worsening of daytime functioning (more daytime impairment symptoms) after treatment compared with before treatment, while a decrease shows an improvement in daytime functioning (fewer daytime impairment symptoms).

**Table 5 | The effect of non-benzodiazepine receptor agonists on daytime impairment symptoms by domain and ontology**

| Endpoint Ontology | Number of daytime impairment symptoms [events per 100 patient years][a] | | | Percentage increase in endpoint after treatment ± SD (*P*-value)[b] |
|---|---|---|---|---|
| | **Before treatment** | **During treatment** | **Average treatment effect [95% CI]** | |
| **All domains** | | | | |
| ICD-clin | 16.1 | 23.2 | 7.0 [−3.2, 17.3] | **43.5% ± 253.9% (*P* = 0.002)** |
| ICD-clin-DiSMOL | 56.0 | 84.0 | 28.4 [13.1, 43.7] | **50.1% ± 377.8% (*P* < 0.001)** |
| DiSMOL | 129.1 | 149.8 | 20.7 [2.6, 38.9] | **16.0% ± 449.3% (*P* = 0.001)** |
| **Cognition domain** | | | | |
| ICD-clin | 16.1 | 23.2 | 7.0 [−3.2, 17.3] | **43.5% ± 253.9% (*P* = 0.002)** |
| ICD-clin-DiSMOL | 30.7 | 38.6 | 8.2 [−3.2, 19.5] | **25.5% ± 280.1% (*P* = 0.008)** |
| DiSMOL | 89.4 | 103.8 | 14.4 [0.2, 28.6] | **16.1% ± 352.0% (*P* = 0.003)** |
| **Emotional domain** | | | | |
| ICD-clin | 15.9 | 22.7 | 6.8 [−3.4, 17.1] | **43.0% ± 253.5% (*P* = 0.002)** |
| ICD-clin-DiSMOL | 29.7 | 51.4 | 21.1 [11.0, 31.1] | **72.9% ± 249.9% (*P* < 0.001)** |
| DiSMOL | 78.7 | 95.7 | 17.0 [2.8, 31.2] | **21.6% ± 352.1% (*P* = 0.001)** |
| **Physical domain** | | | | |
| ICD-clin | 16.0 | 22.9 | 6.9 [−3.3, 17.1] | **43.2% ± 253.6% (*P* = 0.002)** |
| ICD-clin-DiSMOL | 20.7 | 25.6 | 4.9 [−5.7, 15.5] | **23.6% ± 260.3% (*P* = 0.036)** |
| DiSMOL | 105.8 | 118.4 | 12.5 [−3.4, 28.5] | **11.8% ± 395.3% (*P* = 0.016)** |

None-benzodiazepines were found to significantly increase daytime impairment symptoms across all domains (*P*-values ≤ 0.016) when assessed using DiSMOL. When evaluated with ICD-clin and ICD-clin-DiSMOL, non-benzodiazepines were associated with a significant worsening of symptoms across cognitive, emotional, and physical domains (*P*-values ≤ 0.002). However, impairment increases reported by ICD-clin and ICD-clin-DiSMOL were substantially higher than those observed with DiSMOL alone.
*CI* confidence interval, *DiSMOL* Disease-Specific Medical Ontology Learning, *ICD* International Classification of Diseases, *ns* not significant, *SD* standard deviation.
[a]The follow up period was from 182 days before treatment to approximately 3 months (mean) during treatment.
[b]An increase shows worsening of daytime functioning (more daytime impairment symptoms) after treatment compared with before treatment, while a decrease shows an improvement in daytime functioning (fewer daytime impairment symptoms).

treatments could potentially undermine the within-subject analysis' effect estimation validity. The lack of clinical studies for anchoring results also deserves mention.

In conclusion, the DiSMOL framework offers to the best of our knowledge a new approach for learning disease-specific ontologies from physician notes. DiSMOL outperformed the insomnia expert selected ICD-10 codes, in discriminating patients suffering from daytime impairments. Moreover, DiSMOL revealed differences in daytime impairment symptom representation after treatment, highlighting the importance of disease-specific ontologies, particularly in sleep disorders. Embracing real-world data-driven approaches, exemplified by DiSMOL, holds the potential to enhance our comprehension of disease burden and treatment effectiveness. This, in turn, may empower the shaping of evidence-based policymaking and ultimately improve patient outcomes.

## Data availability

Restrictions apply to the general availability of the data because of patient agreements and the nature of patient data. Data used in this manuscript were acquired from HealthVerity for scientific research purposes. Requests for data sharing, of any level, can be directed to clinical-trials-disclosure@idorsia.com for medical and scientific evaluation on a case by case basis, in coordination with HealthVerity. The source data for Fig. 3 is specified within the figure.

## Code availability

Custom code was made for the analysis, which cannot be shared for proprietary reasons. The code is owned by Idorsia.

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

## Acknowledgements

We would like to thank Andrea Phillips Beyer for her insights into the development of IDSIQ, Paul Saskin for helpful discussions on data interpretation, Professor Karsten Borgwardt for giving feedback on our approach, Tobia Ochsner for his contributions to our codebase, and Nikola Cihoric for the engaging discussion around the limitations of current ontologies and for opening our eyes to the fundamental problem of structured healthcare data reporting. We are indebted to Pedro Pina and Christopher Lettieri for their medical advice. The authors further thank Christophe Segalini and Pauline Meijer for their continuing contribution to this project. Medical writing assistance was provided by Melanie Gatt (PhD) and Carlotta Foletti.

## Author contributions

A.B. designed DiSMOL and led the study. R.D., M.F., and M.T. implemented and validated DiSMOL. A.O., M.G., and W.M provided consultation on the research project and interpretation of the results.

## Competing interests

The authors declare the following competing interests: A.B. and A.O. are employees of Idorsia Pharmaceuticals Ltd. R.D., M.F. and M.T. are employees of Visium. M.G. is affiliated with Kemin Foods, CeraZ Technology, and Jazz Pharmaceuticals; Fitbit, Natrol, Merck, Idorsia Pharmaceuticals Ltd, Athleta, and Casper, Maricopa County, NightFood; consulting relationship with Idorsia Pharmaceuticals Ltd; W.M reports being a scientific advisor for Idorsia Pharmaceuticals Ltd. and Carelon.
