## [Transparent Peer Review file · Communications Medicine]

Reviewers' comments:

Reviewer #1 (Remarks to the Author):

The paper presents a novel ontology learning framework for determining disease representations from clinical noting. In particular the approach extends an existing representation of a disease with real-world data extracted from notes, using a vector learning approach augmented by expert validation. I think the idea is good, and interesting, and seems to yield interesting results. The paper is overall well-written and the evaluation is appropriate.

Conceptually I worry that the task is not really an ontology learning or enrichment task, but one of more generally 'learning a disease representation.' This is bolstered by the fact that the starting 'simple' ontology is more of a vocabulary, lacking structure or axiomatisation except for the experts determining three domains of symptoms. The consideration of this instead as a deep phenotyping-type task would enable comparison with a database of disease-phenotype associations such as PheneBank, and may be more appropriate in the context of an approach that does not attempt to learn structure or axiomatisation of the disease representation, as in this case.

I don't think this derails the paper, however, especially in the context of the successful summative study that proves the usefulness of the representation, though I think this consideration could be taken into account by the authors for future studies.

I would like to see a bit more discussion of the relatively low performance of the hypothesised terms - around 1/4th were valid. This can also be considered as a limitation and area for future improvement. What was the content of the discarded terms, were they related at all? Future areas for consideration could be the use of medical BERT embeddings learned from e.g. MIMIC. Otherwise, I would recommend the paper for publication.

Reviewer #2 (Remarks to the Author):

The authors present a report on how they developed a method to increase the sensitivity in identifying daytime impairment in patients with insomnia disorder. It was based on machine learning techniques for natural language processing and derived its raw material from a repository of clinical visit notes by medical doctors. As the starting point for

development, answers to the 14-item Insomnia Daytime Symptoms and Impacts Questionnaire were complemented with terms provided by two insomnia experts to formulate the disease-specific language model. Finally, 425 terms were returned by the language model, and 109 of them were validated by the two experts.

In general, this report provides a tool of clinical relevance and importance for assessment of daytime functioning in patients with insomnia disorder. The report is well written and easy to follow. However, there are points that need revision as follows.

Rows 68-74. Two insomnia experts provided terms of daytime impairment commonly used in their clinical practice to describe the symptoms reported by the patients. Please report how many terms were provided by these two insomnia experts.

Rows 78-81 and 370-372. A model training set comprised of 82,672 patients, whereas a test set comprised of only 50 patients due to the effort involved in understanding the clinical history. However, was the test set big enough? Was the size of it based on convenience only, or was it backed up with any statistical power calculation? Please clarify and report.

Rows 379-382. Daytime impairment during medication with z-drugs, trazodone, and benzodiazepines were analyzed. Were these psychotropics the most widely used ones and therefore selected for analysis, or what was the rationale for selection? Please explain and report. Why was not melatonin included? Please clarify.

Title: Novel medical ontology learning framework to investigate daytime impairment in insomnia disorder and the effect of its treatments

Journal: Communications Medicine

Authors: Alexander J. Büsser, Renato Durrer, Moritz Freidank, Matteo Togninalli, Antonio Olivieri, Michael A. Grandner, William V. McCall

Corresponding author: Alexander J. Büsser email: alexander.buesser@bluewin.ch

Comments	Responses
General	
We thank the Editor for considering our manuscript for submission to Communications Medicine . In support of this submission, we have responded to the comments of the reviewers and revised the manuscript accordingly.	
Reviewer #1	
Conceptually I worry that the task is not really an ontology learning or enrichment task, but one of more generally ‘learning a disease representation.’ This is bolstered by the fact that the starting ‘simple’ ontology is more of a vocabulary, lacking structure or axiomatisation except for the experts determining three domains of symptoms. The consideration of this instead as a deep phenotyping-type task would enable comparison with a database of disease-phenotype associations such as PheneBank, and may be more appropriate in the context of an approach that does not attempt to learn structure or axiomatisation of the disease representation, as in this case. I don’t think this derails the paper, however, especially in the context of the successful summative study that proves the usefulness of the representation, though I think this consideration could be taken into account by the authors for future studies.	Response: Thank you for your insightful observations on our manuscript. We appreciate your perspective that our task might be more closely aligned with deep phenotyping rather than traditional ontology learning or enrichment. We agree that our approach involves significant elements of deep phenotyping, particularly in how we systematically characterize and learn about disease symptoms. However, it is also crucial to recognize that our work extends beyond mere symptom documentation. By incorporating these symptoms into an existing knowledge base, we essentially engage in a form of ontology learning. Our task involves expanding and refining this knowledge base, which already has a specific structural framework—an ontology in its own right. Thus, while deep phenotyping captures the detailed data, our methodology also integrates this data into a structured ontological framework, facilitating better understanding and usage of this information.

	Regarding your suggestion to compare our results with PheneBank, we see the value in such comparisons to benchmark our representations. However, we propose that detailed comparisons of disease representations with external databases like PheneBank would be more appropriate as a focus for future work. This approach will allow us to dedicate the necessary resources to ensure a thorough analysis and meaningful discussion of these comparisons. We believe that framing our work within both the contexts of deep phenotyping and ontology enhancement provides a comprehensive understanding of our contributions to disease representation. We will clarify this dual perspective in our revised manuscript to better communicate the scope and impact of our research.
I would like to see a bit more discussion of the relatively low performance of the hypothesised terms - around 1/4th were valid. This can also be considered as a limitation and area for future improvement. What was the content of the discarded terms, were they related at all? Future areas for consideration could be the use of medical BERT embeddings learned from e.g. MIMIC. Otherwise, I would recommend the paper for publication.	Response: We acknowledge that the validation rate of these terms was approximately 25%, which indeed suggests room for improvement and further refinement in our approach. To clarify, the system was not designed for fully automated use; instead, it was developed to aid experts by providing a broader array of potential terms for further scrutiny. In designing the system, we prioritized sensitivity over specificity to ensure that no potentially relevant terms were prematurely excluded. This approach was intended to supply our domain experts with a comprehensive set of terms, acknowledging that this would require further refinement through human review. Regarding the terms that were ultimately discarded, our analysis shows that while they did not meet the criteria for inclusion in the final dataset, many were loosely related to the domain

	of interest but lacked the necessary specificity or direct relevance to daytime impairment as defined for our study. This observation underscores a potential limitation in our current methodology, which prioritizes a wide retrieval net over the precision of term selection. For future iterations of this research, and especially in scenarios considering a fully autonomous system, we are considering adjusting the thresholds within the percentile of cosine distribution to balance sensitivity and specificity more effectively. Experimenting with different levels of specificity could refine the system’s ability to discern relevant terms without extensive expert intervention. We have added these considerations to the discussion section of our manuscript, noting them as limitations and areas for future research. Additionally, we appreciate your suggestion regarding the use of medical BERT. It's worth noting that in our experiments, we benchmarked our model against various BERT versions and found that BERT models underperformed in identifying known insomnia relationships (see Supplementary Table 5: Model results)
--	--

Reviewer #2

Rows 68-74. Two insomnia experts provided terms of daytime impairment commonly used in their clinical practice to describe the symptoms reported by the patients. Please report how many terms were provided by these two insomnia experts.	Response: Thank you for your question. The terms related to daytime impairment are listed in the supplementary materials of our paper. Specifically, you can find these terms in Supplementary Table 1, which also serves as the model input and is referred to as a simple ontology. In total, 50 unique terms were provided, with some terms being assigned to multiple items and domains, resulting in 55 assignments.
--	--

Rows 78-81 and 370-372. A model training set comprised of 82,672 patients, whereas a test set comprised of only 50 patients due to the effort involved in understanding the clinical history. However, was the test set big enough? Was the size of it based on convenience only, or was it backed up with any statistical power calculation? Please clarify and report.	Response: Thank you for your thoughtful inquiry regarding the size of our test set. The primary reason for the significant discrepancy in size between our training set, which included 82,672 patients, and our test set, which comprised only 50 patients, was due to the intensive effort required in reviewing and understanding detailed clinical histories for the purposes of testing. Each patient's history needed to be thoroughly evaluated by experts to ensure accurate representation and alignment with the model's output, making the process both time-consuming and labour-intensive. Given the constraints associated with this detailed review process, the size of the test set was indeed influenced more by practical considerations of feasibility rather than by formal power calculations. While the test set size might seem small in a conventional statistical context, it was deemed sufficient for the preliminary validation of our model's capability to generalize to new, unseen cases under the rigorous scrutiny of expert review. The insights gained from this focused and detailed examination of 50 cases provided valuable indications of the model's performance and areas for improvement. We acknowledge that future work could benefit from a larger test set, possibly facilitated by streamlined processes or automated tools to reduce the burden of manual review. Expanding the test set would allow for more robust statistical analysis and could help further refine the model's accuracy and generalizability. We have amended the manuscript by noting this limitation.
---	---

Rows 379-382. Daytime impairment during medication with z-drugs, trazodone, and benzodiazepines were analyzed. Were these psychotropics the most widely used ones and therefore selected for analysis, or what was the rationale for selection? Please explain and report. Why was not melatonin included? Please clarify.	Response: Indeed these are the most widely used sleep medications which also have Adverse Events that might interfere with the daytime functioning of the patients. We did not include melatonin as it is not always easy to understand which specific melatonin product is used and OTC formulations may not always be reported. We have amended the manuscript accordingly to address your request.

REVIEWERS' COMMENTS:

Reviewer #1 (Remarks to the Author):

Authors have sufficiently addressed my comments.

Reviewer #2 (Remarks to the Author):

The authors have been receptive, responded to the reviewer comments, and revised their manuscript. I have no further comments to give.